# Bioinformatic Analysis from a Descriptive Profile of miRNAs in Chronic Migraine

**DOI:** 10.3390/ijms251910491

**Published:** 2024-09-29

**Authors:** Alvaro Jovanny Tovar-Cuevas, Roberto Carlos Rosales Gómez, Beatriz Teresita Martín-Márquez, Nathan Alejandro Peña Dueñas, Flavio Sandoval-García, Milton Omar Guzmán Ornelas, Mariana Chávez Tostado, Diana Mercedes Hernández Corona, Fernanda-Isadora Corona Meraz

**Affiliations:** 1Centro de Investigación Multidisciplinario en Salud, Departamento de Ciencias Biomédicas, Centro Universitario de Tonalá, Universidad de Guadalajara, Tonalá 45425, Mexico; alvaro.tovar@academicos.udg.mx (A.J.T.-C.); roberto.rosales@academicos.udg.mx (R.C.R.G.); nathan.pena2658@alumnos.udg.mx (N.A.P.D.); milton.guzman@academicos.udg.mx (M.O.G.O.); mariana.chavez1565@academicos.udg.mx (M.C.T.); diana.hcorona@academicos.udg.mx (D.M.H.C.); 2Instituto de Investigación en Reumatología y del Sistema Músculo Esquelético (IIRSME), Departamento de Biología Molecular y Genómica, Centro Universitario de Ciencias de la Salud, Universidad de Guadalajara, Guadalajara 44340, Mexico; beatriz.martin@academicos.udg.mx (B.T.M.-M.); flavio.sandoval@academicos.udg.mx (F.S.-G.); 3Departamento de Neurociencias, Centro Universitario de Ciencias de la Salud, Guadalajara 44340, Mexico; 4Cuerpo Académico UDG-CA-1096, “Ciencias de la Nutrición y Procesos Moleculares del Metabolismo”, Centro Universitario de Tonalá, Universidad de Guadalajara, Tonalá 45625, Mexico

**Keywords:** microRNAs, miRs, bioinformatic analysis, chronic migraine

## Abstract

Chronic migraines have been described chiefly only from a clinical perspective. However, searching for reliable molecular markers has allowed for the discovery of the expression of different genes mainly associated with inflammation, neuro-vascularization, and pain-related pathways. The interest in microRNAs (miRs) that can regulate the expression of these genes has gained significant relevance since multiple miRs could play a key role in regulating these events. In this study, miRs were searched in samples from patients with chronic migraine, and the inclusion criteria were carefully reviewed. Different bioinformatic tools, such as miRbase, targetscan, miRPath, tissue atlas, and miR2Disease, were used to analyze the samples. Our findings revealed that some of the miRs were expressed more (miR-197, miR-101, miR-92a, miR-375, and miR-146b) and less (miR-133a/b, miR-134, miR-195, and miR-340) than others. We concluded that, during chronic migraine, common pathways, such as inflammation, vascularization, neurodevelopment, nociceptive pain, and pharmacological resistance, were associated with this disease.

## 1. Introduction

Migraine is one of the most prevalent and disabling neurological events that affects mainly working adults, as it affects approximately 14% of the global population [1]. Even though the migraine pathophysiology involves different factors, a significant part of this event is related to the expression of genes associated with neuroinflammation and vascularization, for example, neurovascular disorders that include meningeal vasodilation, oedema formation activation, and sensitization of the trigeminal pain pathways [2].

Migraine is also classified by the frequency of events as episodic or chronic or by the presence or absence of aura, which makes it a complex pathology to study because, just as there are parallels, migraine can behave differently depending on the clinical manifestations of the patients [3].

Chronic migraine is characterized when the episodes occur with a frequency of fifteen times or more per month and has been related closely to epigenetic mechanisms [4]. On the other hand, the diagnosis is merely clinical, and in some cases, imaging studies, such as magnetic resonance, computed tomography, or X-rays, are used with the intention to find any neurological problem, disregarding that a genetic marker could be useful, specific, and reliable.

MicroRNAs (miRs) are non-coding RNAs critical in gene expression regulation. Most of them are transcribed from DNA into primary miRs and processed into precursor miRs and mature miRs. In plenty of cases, miRNAs relate with the 3′ untranslated region (3′ UTR) of target mRNAs, inducing mRNA degradation and translational repression.

MiRs are master regulators of gene expression, inhibiting the translation from not just one but thousands of genes, i.e., a single miR can be a regulator of thousands of genes at once, so the effect of several miRs could be controlling complex signaling processes [5]. Some miRs have been studied in pathways associated with migraine pain, where neurological and immune alterations in the nervous system are the key to pathogenesis [6].

MiRs are among the most recently described markers widely studied in different pathologies, mainly in cancer. They are generally sensitive and specific since they could be increased or decreased at different stages during the course of a disease or condition and can be modified according to treatments. Their availability, presence or absence in different biofluids, can be a reliable indicator of a biological condition, and their identification uses different techniques, ranging from RT-qPCR, microarrays, RNAseq, or next-generation sequencing [2,7].

For instance, mir-96 and the mir-183 families have been associated with chronic pain in mice and can regulate nociceptive genes, and miR-155 and let-7 g have been associated with endothelial function in migraineurs [2].

The identification of molecular markers in migraine is a field under exploration. Currently, there are not enough markers that reflect the inflammatory or pain status in a chronic migraine event. Although some serum miRNAs have already been reported, such as miR-34a or miR-382, it is still complex to link the presence of these miRs with the status or clinical evolution of each patient with their treatment [8,9,10]. Bioinformatics is a tool that has the potential to expand the understanding of the pathophysiological mechanisms of migraine by identifying genes that are targets of the interaction of miRNAs involved in neuroinflammation pathways.

The purpose of this study was to identify novel miRNAs in clinically diagnosed chronic migraine patients and analyze their relationship with neuroinflammation-related pathways by identifying the target genes of interaction resulting from bioinformatics analysis.

## 2. Results

### 2.1. miRs Profile Expression in Patients with Chronic Migraine

From the 192 miRs analyzed, it was observed that 63 were not expressed. From those that showed expression, they were divided into tertiles, and the miRs that were more or less expressed were selected as follows: hsa-miR-197-3p, hsa-miR-101-3p, hsa-miR-92-3p, hsa-miR-375, and hsa-miR-146b-5p were the ones with more expression, and hsa-miR-133a-5p, hsa-miR-133b, hsa-miR-134, hsa-miR-195-3p, and hsa-miR-340 were the ones with lower expressions than the rest of the miRs analyzed (Figure 1).

### 2.2. miR-Targets Interaction in Patients with Chronic Migraine

Bioinformatic analysis of these miRs was performed to know their possible association with different genes. It was observed that each miR could have a semi or total complementarity with approximately 5000 genes; nevertheless, it was found that, for those that were highly expressed, they could regulate six genes at the same time: Pleckstrin Homology Like Domain Family A Member 1 (*PHLDA1*), Ataxin 1 (*ATXN1*), Peroxisome Proliferator-Activated Receptor Gamma Coactivator 1-Beta (*PPARGC1B*), Abelson Interactor 2 (*ABI2*), Low Density Lipoprotein Receptor Class A Domain Containing 4 (*LDLRAD4*), and Nuclear Fragile X-associated Disorder Interacting Protein 2 (*NUFIP2*). Regarding the slightest expressed miRs, only one gene was found to be regulated by all: Tripartite motif containing 44 (*TRIM44*) (Table 1).

On the other hand, the interaction score from each gene that showed association with all miRs was analyzed. It was observed that the genes with the highest score were *PHLDA1*, *ATXN1*, and *TRIM44* (Figure 2).

### 2.3. Signalling Pathways in miR-Targets Profile in Chronic Migraine

Reviewing the pathways that regulate these genes, it was observed that, basically for most of the times, these miRs were more studied and associated for their role in cancer than in other pathologies; conversely, in many cases, we found that they have been studied in neurologic issues, inflammation, and drug resistance, as in the case of the PHLDA1 gene.

In addition, each selected miR was analyzed individually, both those with higher and lower expression, for their association with different pathways. The results indicated hits with neurological, vascular development, and inflammation pathways (Table 2).

### 2.4. Relationship among Chronic Migraine miRs and Other Pathologies

Conversely, according to the pathologies in which they have been studied, the most expressed miRs, hsa-miR-197-3p and hsa-miR-101-3p, have both been reported for Alzheimer’s disease, while all the others coincide in having been studied in some type of cancer. As for the less expressed ones, miR-133a has been studied in vascular diseases.

## 3. Discussion

Various research groups have widely reviewed the expression of miRs and proposed them as biomarkers for the diagnosis, prognosis, and clinical evolution of different pathologies. miRs are master regulators of gene expression by regulating mRNA expression; their activity at the pre-translational level allows for its inhibition by interacting by partial or total complementarity with multiple mRNA targets. Currently, the most widely used serological marker is calcitonin gene-related peptide (CGRP), and the receptor antagonists or anti-CGRP antibodies have been proposed as new therapeutic agents; at the same time, some cytokines, homocysteine, serotonin, hypocretin-1, and glutamate among others have also shown an important role, but the clinical and scientific community does not entirely accept them [23].

Migraine is the most prevalent neurological disability, and little has been developed in terms of sensitive biomarkers. For this reason, the field of microRNAs is something new for this group of headaches. Currently, the diagnosis of chronic migraine is based on the criteria of the International Classification of Headache Disorders (ICDH-3), where clinical variants are considered, such as the frequency of events per month (≥15 days/month for three months) and presence or absence of aura or response to treatment with triptan or ergotamine [24]. In contrast, the pathophysiology of migraine is very complex and involves different nervous system processes that the differential expression of genes can naturally regulate.

In our study, we took blood plasma samples from female patients with chronic migraine, in which we performed a search sweep of multiple miRs, finding miR-197, miR-101, miR-92, miR-375, and miR-146b more expressed than the rest and finding miR-133a/b, miR-134, miR-195, and miR-340 in smaller quantities. In this way, the found miRNAs in patients with chronic migraine have not yet been reported and have not been reported (except for miR-375 [25,26]) for healthy subjects; hence, this allows for us to identify new markers that should be researched in a study including healthy subjects [10,27,28,29,30].

miRs have been related to inflammatory processes, neuropathic pain, nervous system development, and endothelial function. Specifically, they can regulate communication between immune and nervous system cells. This fact is critical because, consequently, they are regulators of pain signaling and have a role in mediating the effects of drugs, since changes in the expression of miRs have been reported depending on the type of treatment, dose, and time of exposure [31].

In this sense, several studies have been conducted in favor of determining a specific panel that could act as a marker. For example, in a study conducted by Andersen et al., 372 miRs were analyzed during migraine attacks, resulting in the differential expression of miR-34a-5p, miR-29c-5p, miR-382-5p, and miR-26b-3p. They also showed that this dysregulation can remain in pain-free periods, while other studies show no differences [32]. It is worth mentioning that, in another study, the elevation of miR-34a and miR-382 was also observed in patients with chronic migraine and excessive use of medications compared to patients with episodic migraine, whose levels decreased when starting a detox [9]. Therefore, the control of clinical variables and experimental conditions must be described to better determine all cases. 

Similarly, Gallelli et al. evaluated the increase in miR-34a-5p and miR-375 during the ictal period in pediatric patients without aura compared to healthy controls and divided their study group into two subgroups: those receiving non-steroidal anti-inflammatory drugs (NSAIDs) or acetaminophen and those receiving any other treatment to relieve pain. They observed that those receiving NSAIDs had decreased miR levels [6].

In other studies, they do not refer to the quality of the patient selection process, for example, whether they are classified by the presence of sensory symptoms when their last migraine event was or if they are under any medication. This makes the comparison of results complicated due to the complexity of the pathophysiology of migraine. Also, it has been studied from other angles. For example, in a study in migraine patients, miRs capable of regulating genes associated with endothelial dysfunction were evaluated, finding miR-155, miR-126, and let-7 g elevated. This suggests that, in some people, migraines may be associated with cardiovascular events [28].

MiR-92 has been reported as a link to cardiovascular diseases and is being studied in multiple cancers. Its inhibition was suggested to maintain the integrity of endothelial cells as its overexpression promotes the proliferation and migration of smooth muscle vascular cells through Krüppel-like factor 4 (*KLF2/4*) and controls macrophage polarization [33].

Another example could be miR-197, which has been decreased in deep vein thrombosis. It was found that it could regulate the Nuclear factor kappa-light-chain-enhancer of activated B cells (NF-κβ) pathway through the CXCR2/COX2 axis, which consequently affects cell proliferation, angiogenesis, and inflammation mechanisms [34].

During the inflammation process, miR-101 can negatively regulate it through inhibition of Mitogen-activated protein kinase 1 (*MAPK1*) and the NF-κβ pathway [35]. This corroborates what Sun et al. observed by demonstrating that the overexpression of miR-101 reduces the expression of pro-inflammatory cytokines, such as interferon gamma (IFN-γ), interleukin-6 (IL-6), and interleukin-17 (IL-17) A, in systemic lupus erythematosus [36]. Likewise, miR-133 can attenuate the inflammatory response and phagocytosis by targeting interleukin 1 receptor associated kinase 1 (*IRAK1*), a component of the Toll-like receptor (TLR) pathway [37]; in addition, another study observed that its expression correlated with the levels of IL-6, interleukin-8 (IL-8), C-reactive protein (CRP), and tumor necrosis alpha (TNF-α) in patients with acute cerebral infarction [38].

In migraines, the typical mechanism is sensory involvement, which may or may not occur in all patients, and the presence of pain caused by inflammation or damage to some tissue or nerve [39]. The activity of miR-134 is related to nociceptive pain, which is caused by the activation of specialized pain sensors when there is inflammation or damage. miR-134 can also induce neuropathic pain and miR-146a when there is damage or dysfunction of the nervous system by targeting TNF receptor associated factor 6 (TRAF6), improving pain signaling [40,41].

Similarly, Li et al., in 2017, observed that miR-375 was under-expressed in mice with tolerance to opioids, and among its targets is the Janus kinase 2/Signal transducer and activator of transcription 3 (*JAK2*/*STAT3*) pathway, which improves tolerance to opioids [42]. In contrast, miR-195 was under-expressed in the dorsal root ganglion in a mouse inflammation model. This model has Ras-related protein Rab-23 (RAB23) as one of its targets, which inhibits the activation of p38 in the MAPK pathway and relieves chronic nociceptive pain [43]. More studies should likely be conducted to more accurately evaluate the miRs that may or may not be playing a role in the migraine process. Specifically, the goal is to achieve a correlation with differential migraine events, such as chronicity, the appearance of sensory symptoms, or the use of different treatments.

## 4. Materials and Methods

### 4.1. Patient Selection and Blood Sample Collection

This is an analytical and descriptive study in which subjects with a diagnosis of chronic migraine were recruited according to the International Classification of Headache Disorders (ICDH-3) criteria who agreed to participate in the study through informed consent. Only three women met all the pharmacological and clinical requirements when interviewed by a neurologist. This study was carried out with the permission of the Ethics Committee of Health of Antiguo Hospital Civil de Guadalajara “Fray Antonio Alcalde” (Date: 13 November 2023, Decision No: HFG/FAA/CEI-1665/23).

The patients’ ages were in the range from 25 to 45 years old with chronic migraine and mentioned having aura, and the samples were taken during the interictal phase (when there was no pain or premonitory symptoms); in addition, they confirmed not being under any medication treatment of psychotropics, non-steroidal anti-inflammatory drugs (NSAIDs), steroids, antibiotics, CNS depressants, or prescription medications of second and third line for migraines. Also, they were not under contraceptives or during menopause. They did not show central nervous system conditions or autoimmune, rheumatic, metabolic, or infectious diseases.

Peripheral blood samples were taken with the BD Vacutainer^®^ system in EDTA tubes. The samples were centrifuged at 1500 RFCs at room temperature to obtain the plasma, which was stored at −80 °C until use.

### 4.2. Extraction and Reverse Transcription of MicroRNAs from Blood Plasma

After a gradual temperature change during the thawing, the plasma samples were treated with the miRNeasy serum/plasma extraction and purification kit (Qiagen ^TM^, Cat. No. 217184, Hilden, Germany), which combines a phenol/guanidine-based reagent and silica membranes for purification. The samples were stored at −80 °C until reverse transcription.

Reverse transcription was performed with a cDNA synthesis kit, Taqman™ Advanced protocol (Applied Biosystems^TM^, Cat. No. A28007, Foster City, CA, USA), which includes four phases for its preparation: poly A tailing reaction, ligation reaction adapter, reverse transcription, and miR-Amp reaction.

### 4.3. MicroRNA Expression

The cDNA obtained from the reverse transcription was diluted with TE buffer to fill the microfluidic cards containing different microRNAs: miRNA Human A (Thermo Fisher^TM^ Cat. No. A31800, Waltham, MA, USA) with the TaqMan™ Fast Advanced Master Mix (Applied Biosystems^TM^, Cat. No. A25576). The microfluidic cards were run under the following conditions: 95 °C for 20 s, 95 °C for 1 s (40 cycles), and 60 °C for 20 s (40 cycles). In total, 192 targets were evaluated, including endogenous and exogenous controls, in the qPCR system QuantStudio from Applied Biosystems^TM^.

### 4.4. Bioinformatic Analysis of the Results

Normalization was performed with hsa-miR-16-5p. From all the miRs tested, 63 of 192 were discarded due to their insignificant expression, leaving only 129 miRs that showed expression. Then, from those 129, only ten were selected because they were found to be more or less expressed within the tertile distribution in the general expression.

Subsequently, these miRs were analyzed in a Python script via the xlsx files corresponding to each miRNA and the genes they had interaction with using the https://www.mirbase.org/ (accessed on 24 January to 10 February 2024) and https://www.targetscan.org/ (accessed on 27–30 January 2024) databases. Duplicates were eliminated, and the gene matching count was performed on those miRNAs.

The genes of interest from all the selected miRs were listed and analyzed according to the interaction score.

Also, each miR was analyzed individually using https://mpd.bioinf.uni-sb.de/ (accessed on 1–5 February 2024, miRPath V2.0) and https://dianalab.e-ce.uth.gr/ (accessed on 1–5 February 2024, miRPath V.3) to examine the signaling pathways involved and https://ccb-web.cs.uni-saarland.de/tissueatlas2/ (accessed on 6–7 February, tissue atlas) to investigate the tissue of major expression. Finally, miR2Disease from https://www.mirbase.org/ (accessed on 24 January to 10 February 2024) was used to explore which kind of pathologies had been studied for these markers.

## 5. Conclusions

Despite the small number of cases, this allows for us to devise new studies by means of bioinformatic analysis that let us clarify the role of miRNAs in the pathophysiological process at comparing them with the different types of migraine, their related pathways, and the changes in the brain’s neurobiology. Importantly, we observed that trigeminal vascular network inflammation is essential in the regulation of the pain processes and the clinical picture in the migraine patient. However, the specific mechanisms of the factors involved in this process are still in the research process. MicroRNAs could explain a little about the relationship of genes associated with inflammation and vascularization as well as their role in chronic pain and the mechanisms of tolerance to analgesics.

## Figures and Tables

**Figure 1 ijms-25-10491-f001:**
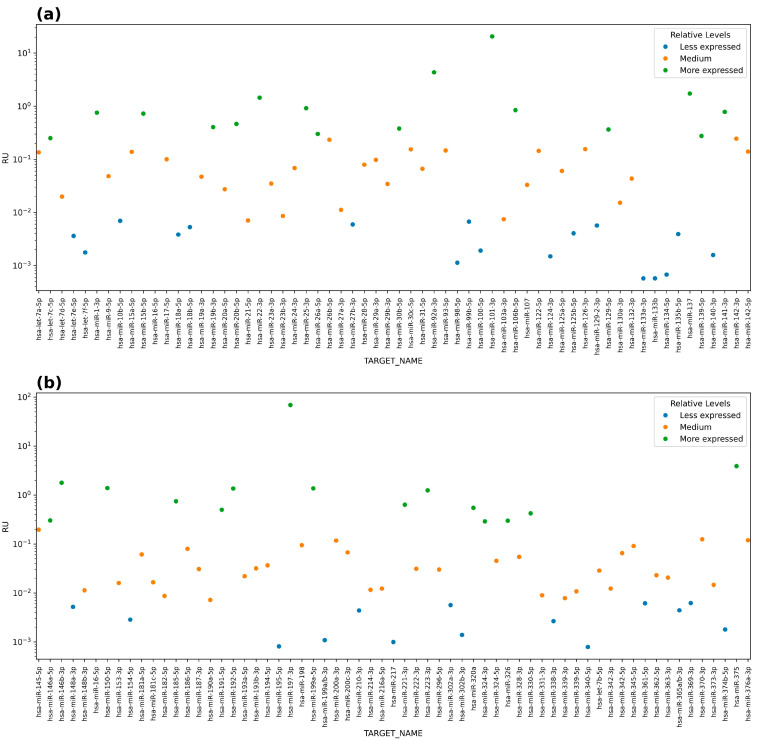
Expression levels of miRNAs profile in chronic migraine. Panels (**a**) and (**b**) show the values in relative expression units (RU); all expression values were organized by tertiles, with those that were most expressed in green and those that were found in lesser amounts in blue; the values that were in the second tertile were arranged in orange.

**Figure 2 ijms-25-10491-f002:**
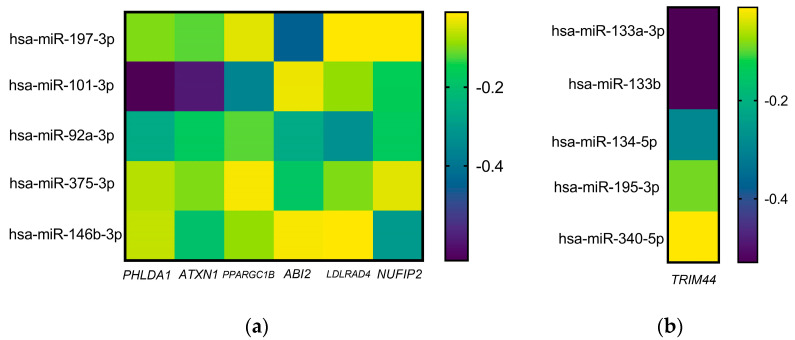
Interaction scores from genes of high coincidence with the highest and lowest expressed miRs: (**a**) Target gene score in the highest expressed miRs, and (**b**) target gene score in the lowest expressed miRs.

**Table 1 ijms-25-10491-t001:** Over- and under-expressed target genes mostly inhibited by miRs.

Gene	Associated Conditions	Regulator miRs
*PHLDA1*	Neuroinflammation [11], Drug resistance [12]	197-3p, 101-3p, 92-3p, 375, 146b-3p
*ATXN1*	Spinocerebellar ataxia type 1 [13] Inflammation via NF-κB regulation [14]	197-3p, 101-3p, 92-3p, 375, 146b-3p
*PPARGC1B*	Gouty inflammation by NLRP3-inflammasome [15]	197-3p, 101-3p, 92-3p, 375, 146b-3p
*ABI2*	Tumor promotion and invasion [16,17]	197-3p, 101-3p, 92-3p, 375, 146b-3p
*LDLRAD4*	Metastasis [18], Intracerebral hemorrhage [19]	197-3p, 101-3p, 92-3p, 375, 146b-3p
*NUFIP2*	Lysosomal damage [20], Neuroinflammation [21]	197-3p, 101-3p, 92-3p, 375, 146b-3p
*TRIM44*	Inflammation [22]	133a, 133b, 134, 195-3p, 340

The target genes inhibited by miRs shown are over- and under-expressed. Abbreviations: *PHLDA1*: Pleckstrin Homology Like Domain Family A Member 1; *ATXN1*: Ataxin 1; NF-κB: Nuclear factor kappa-light-chain-enhancer of activated B cells; *PPARGC1B:* Peroxisome Proliferator-Activated Receptor Gamma Coactivator 1-Beta; NLRP-3: NOD-, LRR-, and pyrin domain-containing protein 3; ABI2: Abelson Interactor 2; *LDLRAD4*: Low Density Lipoprotein Receptor Class A Domain Containing 4; *NUFIP2*: Nuclear Fragile X-associated Disorder Interacting Protein 2; *TRIM44*: Tripartite motif containing 44.

**Table 2 ijms-25-10491-t002:** Neurological, vascularization, and immune-related pathways regulated by each miR.

miRs	Pathway	Hits (P)
hsa-miR-197-3p	Nervous system development	970 (6.45 × 10^−6^)
	Neuronal differentiation	552 (0.001)
	Neuronal projection	477 (0.001)
	Neuronal system	191 (0.002)
	Neurogenesis	645 (0.002)
hsa-miR-101-3p	Nervous system development	950 (9.43 × 10^−7^)
	Neuronal differentiation	441 (1.22 × 10^−4^)
	Neuronal projection	208 (9.80 × 10^−6^)
	Neurogenesis	645 (1.67 × 10^−5^)
	Immune system processes	100 (1.12 × 10^−5^)
	Central nervous system vasculogenesis	2 (0.023)
hsa-miR-92a-3p	Nervous system development	883 (4.12 × 10^−5^)
	Neuronal differentiation	509 (2.87 × 10^−4^)
	Neuronal projection	430 (0.010)
	Neurogenesis	609 (4.23 × 10^−5^)
	Immune system processes	258 (0.023)
	Blood vessel development	277 (0.009)
hsa-miR-375-3p	Nervous system development	792 (3.23 × 10^−7^)
	Neuronal differentiation	441 (9.59 × 10^−4^)
	Neurogenesis	523 (5.37 × 10^−4^)
	Blood vessel development	239 (0.025)
hsa-miR-146b-3p	Nervous system development	886 (1.36 × 10^−4^)
	Neuronal differentiation	513 (4.34 × 10^−4^)
	Neuronal projection	456 (7.68 × 10^−5^)
	Neurogenesis	612 (1.36 × 10^−4^)
	Blood vessel morphogenesis	242 (0.028)
hsa-miR-133a-3p	Nervous system development	745 (1.01 × 10^−7^)
	Neuronal differentiation	424 (6.55 × 10^−5^)
	Neuronal projection	360 (9.13 × 10^−4^)
	Neurogenesis	502 (1.89 × 10^−5^)
	Immune system development	19 (0.003)
	Immune system processes	34 (0.009)
	Vascular development	12 (0.031)
	Blood vessel morphogenesis	12 (0.025)
hsa-miR-133b	Nervous system development	745 (1.01 × 10^−7^)
	Neuronal differentiation	424 (6.55 × 10^−5^)
	Neuronal projection	360 (9.13 × 10^−4^)
	Neurogenesis	14 (0.026)
	Immune system development	12 (0.007)
	Immune system processes	22 (0.021)
	Vascular development	12 (9.29 × 10^−4^)
	Blood vessel development	12 (7.13 × 10^−4^)
hsa-miR-134-5p	Nervous system development	773 (0.002)
	Neuronal projection	402 (1.77 × 10^−4^)
	Neurogenesis	512 (0.036)
	Vascular development	261 (0.007)
	Blood vessel development	250 (0.011)
	Blood vessel morphogenesis	223 (0.009)
hsa-miR-195-3p	Nervous system development	959 (2.50 × 10^−11^)
	Neuronal differentiation	544 (3.35 × 10^−7^)
	Neuronal projection	445 (0.010)
	Neurogenesis	637 (5.54 × 10^−7^)
	Immune system development	380 (0.009)
	Blood vessel development	304 (7.85 × 10^−6^)
	Blood vessel morphogenesis	267 (3.10 × 10^−5^)
hsa-miR-340-5p	Nervous system development	701 (1.02 × 10^−4^)
	Neuronal differentiation	695 (6.98 × 10^−7^)
	Neurogenesis	826 (1.10 × 10^−7^)
	Immune system development	505 (5.11 × 10^−4^)
	Cytokine signaling	8 (0.009)
	Vascular development	226 (0.027)

The evidence of involvement in the pathways presented is by predicted or experimental binding.

## Data Availability

Additional information from the data analysis is available on a drive account that the first author has published. https://drive.google.com/drive/folders/1awptshhZlNcEghC10Lmg51qSX1eHwQuI?usp=sharing.

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
