# Peer review of "Bioinformatic Analysis from a Descriptive Profile of miRNAs in Chronic Migraine"

_ijms, 2024, doi:10.3390/ijms251910491_

Round 1

Reviewer 1 Report

Comments and Suggestions for Authors

Comments and Suggestions:

Title: Bioinformatic analysis from a descriptive profile of miRNAs in chronic migraine.

Reviewer’s report:

The study presented by the authors describes about the identification of deregulated biomarkers related to chronic migraine. They identified miRs that were upregulated (miR-197, miR-101, miR-92a, miR-375 and miR-146b) and downregulated (miR-133a/b, miR-134, miR-195 and miR-340). They finally concluded that, during chronic migraine, common pathways such as inflammation, vascularization, neurodevelopment, nociceptive pain, and pharmacological resistance were associated with chronic migraine.

MicroRNA profiling is a perspective way for studying pathophysiology, searching for biomarkers, or discovering potential treatments. Please upload higher resolution images/figures preferably in TIFF format to have a better view to understand it. In my opinion, the work deserves to be accepted by the journal, after the correction of several major and minor points.

Major Points:

1.      Page 2, line 72: It was stated that out of 192 miRs analyzed, only 63 were not taken for further analysis as they don’t show any significant expression, but there is a discrepancy that in page 8, line 247, 66 miRs were discarded due to their insignificant expression. Please clarify?

2.      Figure 1: The figure legends needs to be explained enough to understand it. Please expand the Y-axis title and write the units if any.

3.      Figure 1: Please explain why two plots were made? If they are showing separate results, please write A and B.

4.      Point 2.1 and 4.4: It was mentioned in 4.4 that 10 deregulated miRs were taken for further analysis but in point 2.1, 6 miRs showed upregulating and 5 showed downregulation. Why is this discrepancy in the results and values are not same throughout the manuscript?

5.      References: I suggest adding in the references section with the following papers related to the topic discussed and not cited: PMID: 36050647 and PMID: 35883029

6.      It is not mentioned in the paper that you confirmed your results for differential expression with the RT-PCR in other validation sets, although this method remains the gold standard for gene expression. I would recommend validating your results using the RT-PCR.

Minor Points:

1.      Introduction: The text is not very well explained and the last para should give a brief description of the whole study.

2.      Conclusion: This part is too short and does not describe about the results concluded.

Comments on the Quality of English Language

Language and Grammar: The manuscript needs a thorough review of text for any grammatical errors and ensure that the language used is clear and concise.

Reviewer 2 Report

Comments and Suggestions for Authors

Comments

The authors collected the blood sample from patient and detected the expression levels of miRNA. They further used bioinformatics tools to analyze the characterization of miRNA. They found that some of the miRs were expressed more (miR-197, miR-101, miR-92a, miR-375 and miR-146b) and less (miR-133a/b, miR-134, miR-195 and miR-340) than others. Finally, they draw conclusions that during chronic migraine, common pathways such as inflammation, vascularization, neurodevelopment, nociceptive pain, and pharmacological resistance were associated 32 with this disease. Overall, the critical link in this study is missing. There are some aspects of the study that are not fully explored.

Major concerns:

1. Migraine is a common neurobiological headache disorder. In this study, the authors tested the changes of miRNA in peripheral blood sample. Given the authors found the levels of miRNA changes in migraine patients, the direct evidence between peripheral levels of miRNA and neurobiological changes is not strong. The authors confirm that the levels of miRNA also changes in brain in animals.

2. The description of method is not clear. When did the author collect the blood samples?

3. The control (health population) is missing. How changes of miRNA in health person needs to be done.

Round 2

Reviewer 1 Report

Comments and Suggestions for Authors

Second revision

Major points:

1.      Page 9: line 287-290: The authors have still not correcting the no. of identified miRs as 63 out of 192 were not carry forward for further analysis and 126 miRs showed expression. How come 192-63= 126?

2.      Figure 2: The miRs in fig2a are on y-axis but in fig2b, they are on x-axis, please make it similar.

Minor points:

1.      Introduction: The introduction is still missing last para that should give a brief description of the whole study.

The manuscript deserves to be accepted by the journal, after the correction of few points.

Author Response

Review ijms-3204378

Title: Bioinformatic analysis from a descriptive profile of miRNAs in chronic migraine

Major Points:

  1. Page 9: line 287-290: The authors have still not correcting the no. of identified miRs as 63 out of 192 were not carry forward for further analysis and 126 miRs showed expression. How come 192-63= 126?

Response 1: We thank you for raising awareness on this concern, this has been addressed on lines 267 and 268: “Leaving only 129 miRs that showed expression…” and “from those 129 only ten…”

  1. Figure 2: The miRs in fig2a are on y-axis but in fig2b, they are on x-axis, please make it similar.

Response 2: Thank you for pointing this out, the figures have been modified accordingly.

Minor points:

  1. Introduction: The introduction is still missing last para that should give a brief description of the whole study.

Response 1: We thank you for your attention to this, we have made changes on page 2 from line 72 to 83.

The manuscript deserves to be accepted by the journal, after the correction of few points.

Dear reviewer, we thank you in advance for your time and understanding, we believe that the text has greatly improved thanks to your kind suggestions.

Reviewer 2 Report

Comments and Suggestions for Authors

The authors did not include the control data or analysis. After including the control data, the conclusions will be solid.

Round 3

Reviewer 2 Report

Comments and Suggestions for Authors

It is can be accepted in the present form.